# Red Pill or Blue Pill? Thresholding Strategies for Neural Network Monitoring

## Abstract

With the increasing deployment of neural networks in critical systems, run-time monitoring plays a critical role in rejecting unsafe predictions during inference. Various techniques have emerged to establish rejection scores that aim to maximize the separability between the distributions of safe and unsafe predictions. In most works, the efficacy of these approaches is evaluated using threshold-agnostic metrics, such as the area under the receiver operating characteristic curve. However, in real-world applications, the effectiveness of a monitor also requires identifying a good threshold to transform these scores into meaningful binary decisions. Despite the pivotal importance of threshold optimization in practice, this problem has received little to no attention in the literature. In this work, we address this question by comparing four strategies for constructing threshold optimization datasets, each reflecting a different assumption about the data available for threshold tuning. We present rigorous experiments on various image datasets and conclude that: 1. Knowledge about runtime threats actually impacting the system helps greatly in identifying an optimal threshold. 2. Without this information, relying solely on in-distribution data is advised, as adding unrelated generic threat data produces worse thresholds.

## 1 Introduction

Deep learning has gained traction in safety-critical domains such as surgical robots (Haidegger, 2019), autonomous vehicles (Ferreira et al., 2022), and drone landing (Guerin et al., 2022a). As reliance on neural networks (NN) in these sectors intensifies, the importance of ensuring their safety keeps growing and demands continued research. NN runtime monitoring is a promising direction, seeking to detect unsafe predictions during inference.

Numerous methods have been developed for NN runtime monitoring. They mostly consist of designing a scoring function indicating the level of confidence for a prediction. These scores can be converted to binary classes through thresholding, to decide which predictions to reject. The quality of a monitor is assessed based on its capacity to build score distributions that effectively separate safe and unsafe predictions. To evaluate this distinction, most of the commonly used metrics are threshold-agnostic, representing an average performance of binary classification metrics across a range of threshold values. Examples of popular metrics include the area under the receiver operating characteristic curve (AUROC) and the area under the precision-recall curve (AUPR). While high values of such metrics often suggest the existence of a good threshold, they do not ensure the ease of finding it.

When deploying a monitor in real-world applications, a concrete rejection threshold value must be set to determine accepted and rejected predictions. This threshold is pivotal, as a good monitor with a poorly chosen threshold can still result in an unsafe system. Despite the crucial nature of this aspect, it remains under-explored in current research.

To address this important question, we propose to evaluate four different strategies for threshold optimization and to assess their performance and practical relevance. The tested strategies represent different ways to construct a threshold optimization dataset, which is then used to find thresholds that maximize chosen metrics (Figure 1). By comparing

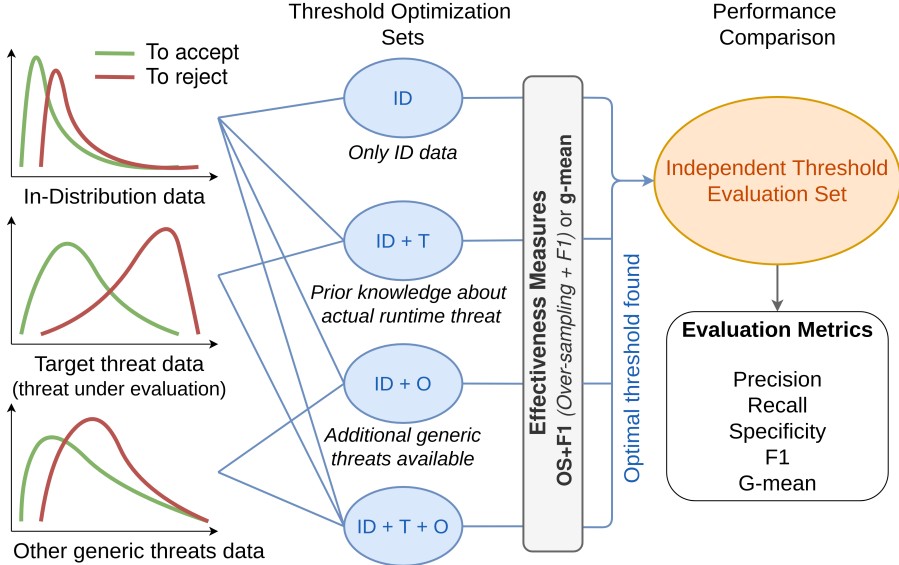

Figure 1: **Conceptual Overview** – This research evaluates four strategies for constructing threshold optimization sets for neural network runtime monitors, each representing distinct assumptions about the data available for threshold tuning.

four threshold optimization strategies, this study aims to answer two important questions: 1. *How beneficial is it to have anticipatory knowledge of probable runtime threats?* 2. *Can we use generic threat sets to facilitate the determination of an optimal threshold?* These questions are answered through extensive experiments, with the objective of providing concrete guidance on this pivotal aspect of neural network monitoring.

This paper is organized as follows: Section 2 reviews relevant literature on NN monitoring and threshold optimization for classification. Section 3 presents our methodology and the associated research questions. Section 4 outlines our experimental design. Section 5 analyzes our findings. In Section 6, we reflect on our findings and explore their practical implications. Finally, in Section 7, we conclude this work and suggest future research directions.

## 2   Background and Related Work

In this section, we lay out key definitions related to NN runtime monitoring, and we present the relevant literature. Although this work focuses on classification, it is worth noting that some of the methodologies discussed here are transferable to other Machine Learning tasks.

### 2.1   Neural Networks Runtime Monitors

Let us denote a classification task by $T$, its feature space by $\mathcal{X}$, and its label space by $\mathcal{Y}$. The oracle function for $T$ is denoted $\Omega$, signifying that the ground truth for any $x \in \mathcal{X}$ is $\Omega(x)$. Let $D_{\text{train}}$ represent a training dataset for $T$, and let $f$ be a classifier for $T$, trained using $D_{\text{train}}$. A runtime monitor for $f$, denoted as $m_f$, is a binary classifier designed to filter out unsafe predictions of $f$. In this work, we adopt a convention where the positive class denotes unsafe samples, though the reverse convention also exists in the literature.

Most of the literature on NN monitoring does not focus on constructing binary classifiers, but rather on developing models that output continuous scores representing the confidence in a prediction. In practice, training a monitor, i.e., adjusting the parameters of the monitoring function to generate meaningful scores, commonly involves the use of the same labeled training dataset, $D_{\text{train}}$, although this is not a strict requirement. Converting these scores into binary classification outputs requires applying a thresholding operation.

The fitting method typically relies on features extracted from one or more layers of the classifier $f$. Hendrycks & Gimpel (2016) proposed to detect abnormal examples using the maximum softmax probability (MSP) as their score. Lee et al. (2018) fitted class-conditional Gaussian distributions to the features extracted and defined their confidence score as the minimum Mahalanobis distance to class-wise centroids. Henzinger et al. (2020) compared novel inputs to the smallest bounding boxes containing features from $D_{\text{train}}$. Liu et al. (2020) proposed the energy score (*logsumexp* of the logits) and Sun et al. (2021) suggested computing rectified logits by clipping the activations. Recently, Wang et al. (2022) developed a score based on virtual logits, generated from the norm of feature residuals against the principal subspace defined by the training set. Next, we present the different perspectives about the crucial topic of NN monitor evaluation.

## 2.2 Evaluation of Neural Networks Runtime Monitors

**Out-Of-Distribution vs. Out-of-Model-Scope** – The concept of *safety* is central to defining runtime monitoring. The existing literature presents two coexisting perspectives regarding what constitutes an unsafe sample (Guerin et al., 2023). The first view targets the detection of Out-of-Distribution (OOD) data, i.e., data that falls beyond the training distribution of the classifier, represented by $D_{\text{train}}$. The second view seeks to identify Out-of-Model-Scope (OMS) data, which triggers incorrect predictions of the classifier. For training, there is no conceptual distinction between OOD and OMS. The evaluation criterion is what primarily distinguishes these settings. As explained by Guerin et al. (2023), by evaluating a monitor's proficiency in rejecting incorrect predictions, the OMS approach circumvents the potentially ambiguous definition of "OODness" and avoids any misconceptions about OOD detection performance. Therefore, in this study, our experiments follow the OMS setting, where we expect a monitor to return 0 when $f(x) = \Omega(x)$ and 1 if $f(x) \neq \Omega(x)$.

**Evaluation Dataset Construction** – In most works, evaluating monitors involves using in-distribution (ID) test data, along with some threat data, to assess the performance outside of the training distribution. For ID test data, we usually use the test split associated with $D_{\text{train}}$. The threats primarily fall into three categories: 1. *Novelty* – The labels do not belong to the label space ($\Omega(x) \notin \mathcal{Y}$), 2. *Covariate Shift* – The inputs are not drawn from the same distribution as $D_{\text{train}}$, 3. *Adversarial Attacks* – The inputs are maliciously modified to cause misclassifications. In the OMS setting, labeled datasets are used so that errors of $f$ can be identified to serve as ground truth for the monitor evaluation. Both the test set and threat sets may contain misclassifications. Additionally, except for novelty, the threat sets can contain correct predictions, depending on the degree of perturbations applied.

**Threshold Agnostic Evaluation Metrics** – A monitor is evaluated based on its ability to distinguish correctly classified data from misclassifications. Related works frequently use threshold-agnostic metrics to assess this skill across a range of thresholds. Examples of such metrics include AUROC, AUPR, and FNR@95TNR (False Negative Rate at 95% True Negative Rate). However, to deploy a runtime monitor in a real-world scenario, one must select a fixed threshold value to decide which predictions to reject. As of today, no studies have specifically addressed this threshold selection mechanism for monitoring. Threshold selection is typically addressed in a somewhat nebulous manner, suggesting that the "threshold should be chosen such that a high proportion of ID data instances are accurately processed by the monitor" (Liu et al., 2020; Sun et al., 2021; Wang et al., 2022).

## 2.3 Threshold optimization for classification

Despite the absence of work addressing threshold fitting for NN runtime monitoring, some research has tackled this problem in the broader context of classification. Arampatzis & van Hameran (2001) explained the steps involved in the exhaustive search method for threshold optimization on a finite test dataset: 1. Calculate the classification scores for all samples of the test dataset, 2. Sort the list of predicted scores, 3. Select a metric to represent threshold performance, called *effectiveness measure*, 4. Calculate the effectiveness measure at every position of the sorted list, 5. Find the position where the effectiveness measure is optimal, 6. Set the threshold between the score corresponding to this position and the next one.

In the literature, the most common variations of this standard optimization pipeline involve alternative choices for the effectiveness measure: F-score (Zou et al., 2016), geometric mean of Recall and Specificity (Johnson & Khoshgoftaar, 2021), Matthews correlation coefficient (Chicco & Jurman, 2023), or Cohen's kappa (Freeman & Moisen, 2008). Another research direction involves developing optimized search strategies to identify the threshold more efficiently (Arampatzis & van Hameran, 2001; Esposito et al., 2021).

In this study, we discuss different threshold optimization strategies for runtime monitors, consisting of different ways to construct the validation set used to optimize the threshold.

## 3 RESEARCH QUESTIONS

Let us consider a monitor $m_f$, that has been trained to produce scores reflecting the confidence of a NN. Our goal is to compare various strategies for determining a threshold for these scores, to determine the predictions to reject. Although the process of finding a suitable threshold has received little attention in the literature, it is a crucial factor to consider. In practice, a monitor may generate scores that accurately distinguish incorrect predictions, but its safety could be compromised if the rejection threshold is not properly calibrated.

To evaluate the effectiveness of a given threshold, we employ conventional binary classification metrics, such as Recall and Precision, on a carefully designed test dataset, which we call *Threshold Evaluation Set*. To have a balanced evaluation, we construct the Threshold Evaluation Set to encompass regular in-distribution data as well as one specific target threat. The inclusion of in-distribution data enables us to identify monitors that may overly reject, and focusing on a single threat allows us to characterize distinct monitor failures. This focus is more realistic, as it is unlikely for a NN to encounter multiple threats concurrently.

To tune the threshold, we use a separate *Threshold Optimization Set*. Fitting the threshold essentially involves identifying the value that optimizes a specific effectiveness measure on the Optimization Set (see Section 2.3). The chosen effectiveness measure should reflect the delicate balance between system safety and availability, i.e., it should encapsulate the monitor's capacity to reject incorrect predictions and to accept correct ones (Guerin et al., 2022b). In our experiments, we try F1 and g-mean (see Section 4). Both the Threshold Optimization and Evaluation sets are composed of inputs to the NN (images), corresponding monitor scores, and labels that indicate the correctness of the predictions.

The four strategies explored in this study consist of distinct methodologies to construct the Optimization Set (Figure 1). They reflect alternative real-world deployment scenarios for monitors, representing assumptions about our ability to anticipate forthcoming threats.

- The first strategy, denoted ID, involves constructing an optimization set composed exclusively of In-Distribution (ID) data samples. This presumes that no threat data is accessible for threshold selection. In the remaining strategies, ID data samples are a common component, but other samples corresponding to threats are incorporated.
- The second strategy, denoted ID+T, involves enriching the optimization set with data samples associated with the Target threat (T), i.e., the threat under evaluation. This scenario corresponds to situations where threats pertinent to the system have been previously identified, such as through a system safety analysis.
- The third strategy, denoted ID+O, designs an optimization set without the target threat, but including samples corresponding to Other generic threats (O). This scenario examines if awareness of generic threats can aid in determining a more effective threshold for unanticipated, new threats.
- Lastly, the fourth strategy, denoted ID+T+O, employs an optimization set containing data samples for both the Target Threat and Other generic threats. It aims to assess the performance of a monitoring threshold when multiple threats are used for fitting, and one of them is the target threat.

A summary of how the Optimization and Evaluation sets are constructed for the different strategies can be found in Table 1. It shows that the Threshold Evaluation Set is always the same and never overlaps with the Optimization set.

Table 1: **Threshold Optimization and Evaluation sets** – Methodology to construct the threshold optimization and evaluation sets for the different strategies considered in this study. Set 1 and Set 2 always denote non-overlapping splits of a dataset.

| | | In-Distribution | | Target Threat | | Other Generic |
|---|---|---|---|---|---|---|
| | | Set 1 | Set 2 | Set 1 | Set 2 | Threats |
| | ID | ✓ | | | | |
| Threshold | ID+T | ✓ | | ✓ | | |
| Optimization | ID+O | ✓ | | | | ✓ |
| | ID+T+O | ✓ | | ✓ | | ✓ |
| Threshold Evaluation | | | ✓ | | ✓ | |

Our experiments within these four strategies aim to address two primary research questions:

RQ1 *How helpful is prior knowledge about the target threat?*

RQ2 *How helpful is the inclusion of generic threats data?*

To answer RQ1, we compare ID against ID+T, and ID+O against ID+T+O. If our findings reveal that prior awareness about the evaluated threat is crucial, it could significantly limit the applicability of runtime monitors. Indeed, the main objective of monitoring is to address unforeseen hazards. If knowledge about the actual threats that an NN will encounter is readily available, such examples would typically be incorporated during training. It is worth noting that several studies have used this strategy for tuning monitor hyperparameters by simply dividing the evaluation set into validation and test subsets (Hsu et al., 2020).

For RQ2, we compare strategy ID against ID+O, as well as ID+T against ID+T+O. A positive answer would be promising, given the relative ease of constructing a generic dataset of threats, which could be utilized to enhance monitoring system performance and subsequently facilitate the adoption of neural networks in safety-critical systems.

## 4 EXPERIMENTAL DESIGN

### 4.1 DATASETS, MODELS AND MONITORS

To answer the aforementioned research questions, we conducted extensive experiments. To encapsulate varying ID scenarios, we use three image classification datasets: CIFAR10, CIFAR100 (Krizhevsky et al., 2009) and SVHN (Netzer et al., 2011). For each ID dataset, we use 2 distinct neural network architectures – DenseNet and ResNet – with weights taken from Lee et al. (2018). For Densenet, the test accuracies are: CIFAR10 (0.93), CIFAR100 (0.73), SVHN (0.88), and for ResNet: CIFAR10 (0.92), CIFAR100 (0.73), SVHN (0.89).

For each ID dataset and architecture pair, we implement four distinct monitoring techniques. Mahalanobis (Maha) (Lee et al., 2018) and Outside-the-Box (OtB) (Henzinger et al., 2020) are feature-based approaches. We derive the feature representation from the final layer preceding classification and do not apply input pre-processing. On the other hand, Max Softmax Probability (MSP) (Hendrycks & Gimpel, 2016) and Energy (Ene) (Liu et al., 2020) are logit-based methods. Regarding hyperparameters, we use num_box=3 for OtB and T=1 for Ene. These settings resulted in a total of 24 monitors evaluated.

Each ID set is paired with nine unique threat sets to assess the monitors under varied circumstances. These threats encompass three novelty sets (datasets with classes distinct from the ID set), three covariate shifts (diverse transformations from AugLy (Papakipos & Bitton, 2022), and three adversarial attacks (generated with Torchattacks (Kim, 2020)). All these threats are parameterized in the same way as Guerin et al. (2023).

### 4.2 THRESHOLD OPTIMIZATION METHODOLOGY

For each ID dataset–monitor pair, we cycle through the nine threats, with each serving once as the target Threat (T), resulting in nine unique outcomes for each strategy. While assessing

a chosen target threat, the remaining eight threats serve as Other Generic Threats (O). The test split of the classifier's training dataset serves as the In-Distribution (ID) dataset. Then, both the ID set and the T set are randomly split in half, so that the threshold evaluation set and the four threshold optimization sets can be constructed, following the methodology presented in Section 3 (Table 1).

To optimize the threshold on the optimization set, we follow the methodology described in Section 2.3. For the effectiveness measure, we initially used F1, the harmonic mean between Precision and Recall, as it is a prevalent choice in the literature. Yet, early experiments revealed that F1 frequently resulted in the unfavorable action of setting exceedingly low thresholds, thereby rejecting all samples in the evaluation set. Of the 864 experiments conducted (3 ID $\times$ 2 NN $\times$ 4 monitoring approaches $\times$ 9 threat sets $\times$ 4 optimization sets), this outcome happened 116 times. Such behavior can be attributed to the significant class imbalance often observed in our optimization sets. Indeed, since classifiers typically commit fewer errors with ID data, the ID strategy predominantly contains negative examples (designated for acceptance), and other strategies, notably ID+O and ID+T+O, contain much more positive examples (designated for rejection).

We tested two distinct solutions to address this challenge: 1. over-sampling (OS) the minority class in the threshold optimization set to achieve a positive-to-negative ratio between 0.4 and 0.6, 2. using another effectiveness measure: g-mean, the geometric mean between Recall and Specificity. As Specificity solely considers samples with negative ground truth, g-mean is unaffected by class imbalance. A very low threshold results in a recall of 1 and a specificity of 0, and will not be favored by g-mean optimization. Both OS+F1 and g-mean approaches are compared in our experiments.

### 4.3 Evaluation Metrics and Statistical Synthesis

Once a threshold is chosen, we evaluate its performance on the threshold evaluation set. For each experiment, we compute five evaluation metrics (F1, g-mean, Recall, Precision, Specificity) representing different aspects of the monitor's performance.

Given the comprehensive scope of our experiments, we are left with 1728 recorded outcomes for each of these five metrics. Drawing definitive conclusions from such an expansive set of raw results is challenging. Even when we fix the effectiveness measure, we are still tasked with comparing the four threshold optimization strategies across 216 cases (864 divided by 4 strategies). Consequently, we resort to statistical testing to discern the distinctions between different approaches across multiple results. Adhering to the methodology outlined by Demšar (2006), we employ the non-parametric Wilcoxon signed-rank tests for comparing two strategies over multiple scenarios. Meanwhile, the Friedman test and its associated Nemenyi post-hoc test are utilized for comparing multiple strategies across multiple scenarios.

## 5 Results

The data from our 1728 experiments is complex and not immediately interpretable in its raw form. In this section, we present the outcomes of our statistical analysis and draw associated conclusions. For transparency and reproducibility, all results detailed in this paper can be replicated using the code available in the supplementary material.

### 5.1 Comparison of effectiveness measures

First, we compare the two proposed effectiveness measures for threshold tuning on the Optimization set: over-sampling with F1 (OS+F1) and g-mean. For each of the 4 strategies and each of the 5 evaluation metrics, we compare these effectiveness measures using the Wilcoxon signed-rank tests across the 216 experiments. The Wilcoxon test is a non-parametric statistical test, used to compare the performance of two classifiers over multiple datasets (Demšar, 2006). The results obtained are shown in Table 2. We find that OS+F1 generally yields better Recall and F1 scores, whereas g-mean optimization produces better Precision, Specificity, and g-mean scores. These findings indicate that the choice of the

Table 2: **Effectiveness measures comparison (OS+F1 vs. g-mean)** – Metrics were computed across the 216 experiments, followed by statistical comparison using the Wilcoxon test. The displayed numbers represent p-values, underlined orange text indicates OS+F1 is worse than g-mean, regular blue text indicates OS+F1 is better than g-mean, and italicized black text indicates no significant difference.

|             | ID    | ID+T  | ID+O  | ID+T+O |
|-------------|-------|-------|-------|--------|
| F1          | 3e-08 | 3e-04 | 2e-04 | 8e-06  |
| G-mean      | 1e-26 | 4e-29 | 2e-02 | *5e-02* |
| Recall      | 4e-31 | 2e-32 | 4e-37 | 4e-37  |
| Precision   | 1e-31 | 1e-32 | 9e-36 | 2e-35  |
| Specificity | 1e-31 | 2e-32 | 3e-37 | 3e-37  |

effectiveness measure should be based on the particular metric one seeks to optimize, and this choice should be aligned with the objectives of the system under test. A higher Recall corresponds to a more conservative system, i.e., fewer false acceptances from the monitor. Conversely, higher Precision and Specificity indicate an improved system availability, i.e., fewer false rejections from the monitor. More results comparing effectiveness measures can be found in Appendix A.1.

## 5.2 COMPARISON OF THRESHOLD OPTIMIZATION SET CONSTRUCTION STRATEGIES

Next, we compare the monitoring performance obtained with the different strategies to construct the Threshold Optimization set. To compare several approaches across multiple experiments, we use the Friedman test and its corresponding Nemenyi post-hoc test, as recommended by Demšar (2006). The Friedman test is a non-parametric test comparing the average ranks of different models, with the null hypothesis assuming no significant difference between them. If the null hypothesis is refuted, the Nemenyi post-hoc test is then applied to identify which model has greater performance.

More precisely, we compare the values obtained for the F1 and g-mean scores on the Threshold Evaluation sets. We focus on these metrics because they are both intended to represent a balance between over-rejection and over-acceptance. At the significance level of $\alpha = 0.05$, the Friedman test shows a significant difference in performance between the four threshold optimization strategies. The results of the Nemenyi test, with both OS+F1 and g-mean as effectiveness measures, are presented in Figure 2. These results allow us to formulate explicit responses to our research questions. We note that results for other evaluation metrics (Recall, Precision, Specificity) are given in Appendix A.2.

*RQ1. How helpful is prior knowledge about the target threat?*

As anticipated, the best strategy is ID+T, where the Optimization set closely mirrors the Evaluation set. Interestingly, the ID+T+O and ID+O strategies consistently demonstrate statistically equivalent performance. This suggests that if one opts to utilize a large set of generic threats for threshold tuning, the inclusion of target threat data becomes useless. This is due to the fact that target threat data samples in the Threshold Optimization set are diluted among the other threats, diminishing their influence on the threshold selected.

*RQ2. How helpful is the inclusion of generic threat data?*

With OS+F1 as the effectiveness measure, the ID+O strategy outperforms ID. Conversely, with g-mean as the effectiveness measure, ID outperforms ID+O. Hence, to know precisely the benefits of incorporating other generic threats, we perform a Wilcoxon test to compare the ID strategy optimized with g-mean to ID+O optimized with OS+F1. Our results reveal that the ID strategy is superior to ID+O when evaluating g-mean scores on the Evaluation sets (p-value=3e-10) and that there is no statistical difference between the two strategies for the F1 evaluation metric (p-value=0.2). In other words, without knowledge about the expected threats a system might face, it is preferable to rely solely on in-distribution data to determine the monitoring threshold and to use g-mean for optimization.

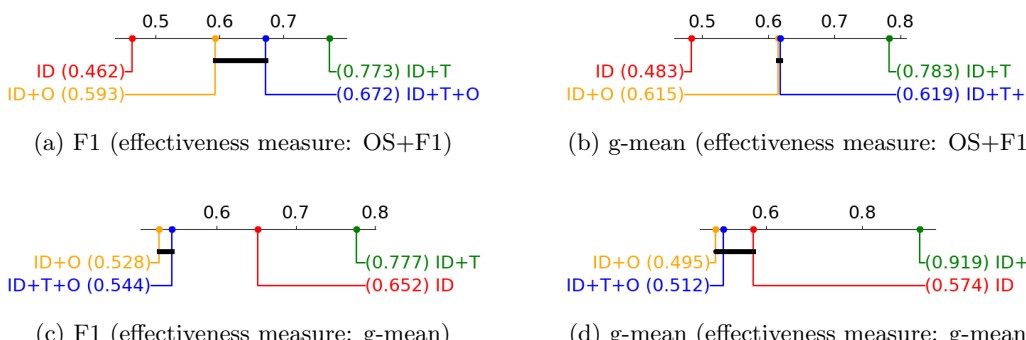

(a) F1 (effectiveness measure: OS+F1)  (b) g-mean (effectiveness measure: OS+F1)

(c) F1 (effectiveness measure: g-mean)  (d) g-mean (effectiveness measure: g-mean)

Figure 2: **Threshold Optimization sets comparison** – Critical distance diagram showing the results of the Nemenyi test. The horizontal axis represents the average rank of the strategies. A black bar connecting two or more strategies indicates no significant difference.

Figure 2 also indicates that the ID+T strategy is better than ID+T+O. This suggests that supplementing the Threshold Optimization set with an arbitrary pool of threat data samples is not beneficial. If the target threat has been identified, it is advisable to use a combination of ID data and data representing this specific threat. Introducing data related to other random threats would simply penalize the monitoring performance.

## 6    UNDERSTANDING THE RESULTS THROUGH AN EXAMPLE

Our experiments confirmed the superiority of the ID+T strategy for building the Threshold Optimization dataset. As anticipated, tuning the threshold with data closely mirroring the evaluation dataset yielded the best results. However, the decreased performance observed when adding generic threats to the Optimization set is less intuitive. In this section, we propose to try understanding this behavior through the study of a specific example.

To ensure that the chosen example offers meaningful insights, we select a case where the performance differences across strategies align with the conclusions presented above. For clear visualization, we require a monitor that exhibits good separability (AUROC > 0.8 on the Evaluation set), and we select the scenario that shows the maximum performance variability among strategies. Details about the chosen example can be found in Appendix A.3. Figure 3 shows the distributions of monitoring scores of the Threshold Optimization sets for the ID, ID+O, and ID+T strategies, as well as for the Threshold Evaluation set. The thresholds derived from both effectiveness measures are also displayed.

Examining Figures 3b and 3d, we observe that the ID+T strategy yields score distributions most resembling those in the Evaluation set, leading to near-optimal thresholds, especially when using g-mean. In contrast, the ID strategy (Figure 3a) shows error scores (in blue) that are too close to the correct ones, resulting in smaller thresholds. However, it is worth noting that the ID strategy performs particularly well for this example, likely due to FGSM attacks generating images closely resembling the originals.

Figure 3c shows the limitations of the ID+O strategy. Interestingly, the failures differ based on the effectiveness measure used. With OS+F1, the threshold is too small because the error score distribution stretches excessively to the left. When optimizing for F1, we try to minimize missed errors, i.e., maximize Recall, thus pushing for a smaller threshold. Conversely, with g-mean, the threshold is set excessively high because the correct score distribution stretches excessively to the right. This is due to g-mean optimization prioritizing reducing false rejections to maintain Specificity. The wide spread in score distributions for ID+O can be attributed to the inclusion of a large variety of threat data, containing a spectrum from correctly classified data deviating from the training distribution to imperceptible threats triggering errors.

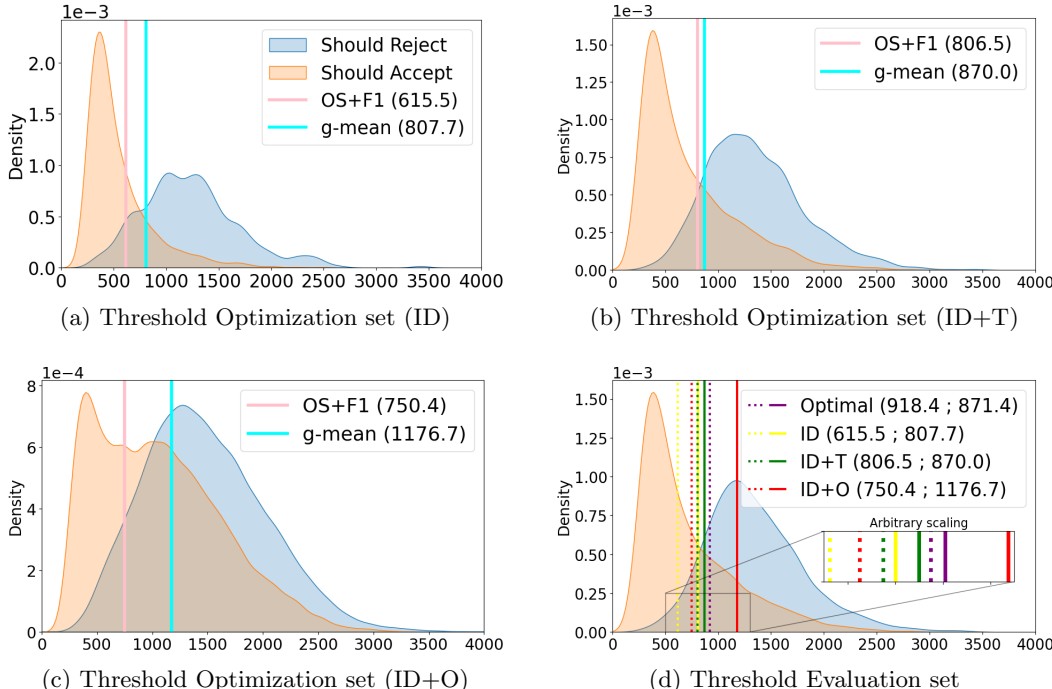

Figure 3: **Visual example to explain our findings** – Distributions of monitoring scores for the Optimization (ID, ID+O, ID+T) and Evaluation sets. Selected example: ID data: CIFAR10, threat: FGSM, NN: Resnet, monitor: Mahalanobis. Vertical lines represent thresholds obtained with different effectiveness measures. In (d), the dashed (resp. plain) lines represent thresholds obtained with OS+F1 (resp. g-mean). The "Optimal" thresholds maximize their respective effectiveness measures on the Evaluation set.

## 7  CONCLUSION

In this study, we compared threshold optimization strategies for runtime monitoring of neural networks. Our findings confirmed the efficacy of the ID+T strategy, which employs data from both the training distribution and the anticipated system threat, in determining the most optimal thresholds. For this scenario, the appropriate effectiveness measure depends on the specific objectives of the monitor. Using F1 with over-sampling results in more conservative monitors, reducing missed errors, while using g-mean encourages higher system availability, reducing false rejections. However, it is not always feasible or appropriate to base monitoring on anticipated system threats. Indeed, monitors are often designed to protect systems against unforeseen threats. In such cases, our study reveals that solely using ID data and adopting g-mean as the effectiveness measure, yields the most effective thresholds.

The example discussed in Section 6 suggests that incorporating data samples from a multitude of unrelated threats results in overly dispersed distributions of correct and error scores. Consequently, this approach does not produce optimal outcomes. Hence, a promising future research direction might be to explore the integration of data samples from more narrowly defined threat categories. This could pave the way for designing monitors specifically tailored for distinct classes of anticipated threats, such as adversarial attacks.

Finally, we also emphasize that our experimental methodology, designed to explore the thresholding challenge for NN monitoring, can be adapted to examine the tuning of other hyperparameters. Many existing studies in the literature often split the evaluation dataset into validation and test sets for parameter optimization, which is equivalent to employing the ID+T strategy. Employing our proposed framework could provide deeper insights into how much existing monitoring techniques rely on prior knowledge of actual threats.

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

# A   Appendix

## A.1   Further comparisons of effectiveness measures

In section 5, we compared two effectiveness measures for threshold tuning: g-mean (geometric mean of Recall and Specificity), and F1 (harmonic mean of Precision and Recall) complemented with over-sampling (OS+F1). Here, we extend this analysis to include F1 without over-sampling, and a typical threshold used in the literature, chosen such that the True Negative rate of the Threshold Optimization set is set to 0.95. We use the same protocol for comparison, employing the Wilcoxon signed-rank test to determine what effectiveness measure yields better performance across five evaluation metrics. We note that the performance is always evaluated on the appropriate Threshold Evaluation sets.

To confirm that the oversampling approach is beneficial, we compare using F1-score with and without oversampling as effectiveness measures. Table 3 shows the results. In general, oversampling gives better F1 and g-mean scores on the Threshold Evaluation set. We can conclude that this oversampling strategy works and reduces the bad behavior of rejecting all inputs in imbalanced scenarios (see Section 4).

Table 3: **Effectiveness measures comparison (F1 with oversampling vs. F1 without oversampling)** – Metrics were computed across the 216 experiments, followed by statistical comparison using the Wilcoxon test. The displayed numbers represent p-values, underlined orange text indicates F1 with oversampling is worse than F1 without oversampling, regular blue text indicates F1 with oversampling is better than F1 without oversampling, and italicized black text indicates no significant difference.

|  | ID | ID+T | ID+O | ID+T+O |
|---|---|---|---|---|
| F1 | *9e-01* | 2e-24 | 3e-12 | 3e-13 |
| G-mean | *6e-02* | 8e-08 | 3e-23 | 6e-26 |
| Recall | 1e-35 | 4e-06 | 7e-18 | 3e-26 |
| Precision | 4e-35 | 1e-08 | 2e-20 | 3e-26 |
| Specificity | 1e-35 | 2e-03 | 3e-22 | 4e-26 |

In the literature, it is common to use FNR@95TNR (False Negative Rate at 95% True Negative Rate) as a monitoring evaluation metric. This means that the threshold is set such that 95% of correct predictions are actually accepted by the monitor. Here, we evaluate this standard literature threshold against the threshold obtained from proper optimization with g-mean as the effectiveness measure. Table 4 clearly shows that threshold optimization is better than 95% TNR for balanced metrics (F1 and g-mean). Precision and Specificity are better for 95% TNR by construction. We also note that similar results were obtained when comparing 95% TNR against OS+F1.

Table 4: **Effectiveness measures comparison (@95TNR vs. g-mean)** – Metrics were computed across the 216 experiments, followed by statistical comparison using the Wilcoxon test. The displayed numbers represent p-values, underlined orange text indicates metric score with threshold chosen @95TNR is worse than optimized with g-mean, regular blue text indicates metric score with threshold chosen @95TNR is better than optimized with g-mean, and italicized black text indicates no significant difference.

|  | ID | ID+T | ID+O | ID+T+O |
|---|---|---|---|---|
| F1 | 6e-16 | 4e-30 | 4e-37 | 4e-37 |
| G-mean | 3e-21 | 3e-37 | 3e-37 | 3e-37 |
| Recall | 3e-37 | 3e-37 | 3e-37 | 3e-37 |
| Precision | 7e-35 | 1e-26 | 1e-25 | 1e-22 |
| Specificity | 3e-37 | 3e-37 | 3e-37 | 3e-37 |

## A.2 Further Comparisons of Optimization Set Construction Strategies

In this section, we present more performance comparisons of the four strategies for constructing the Threshold Optimization set, using other evaluation metrics (Recall, Precision, and Specificity). The results obtained with OS+F1 as the effectiveness measure are given in Figure 4 and the results obtained with g-mean as the effectiveness measure are given in Figure 5.

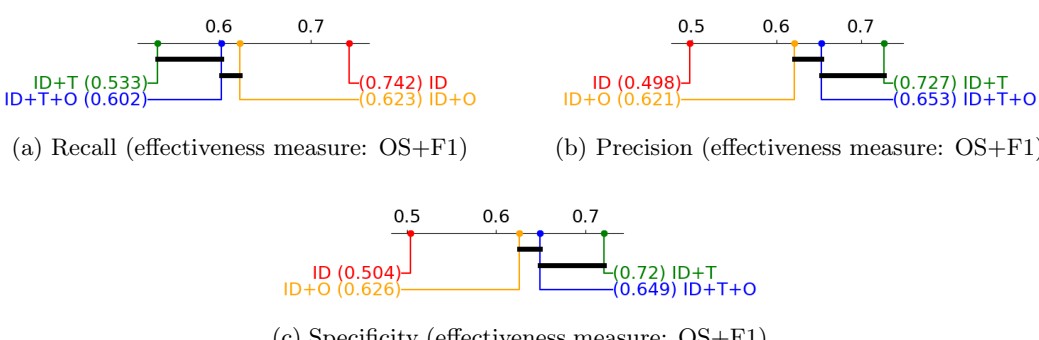

(a) Recall (effectiveness measure: OS+F1)  (b) Precision (effectiveness measure: OS+F1)

(c) Specificity (effectiveness measure: OS+F1)

Figure 4: **Threshold Optimization sets comparison, with OS+F1 as the effectiveness measure** – Critical distance diagram showing the results of the Nemenyi test. The horizontal axis represents the average rank of the strategies. A black bar connecting two or more strategies indicates no significant difference.

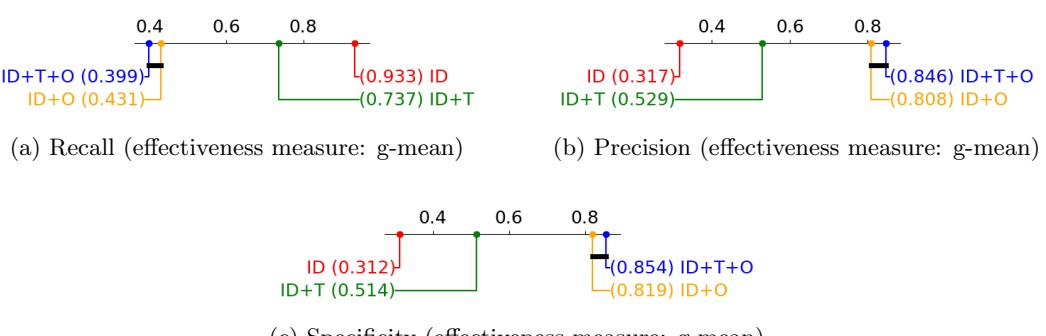

(a) Recall (effectiveness measure: g-mean)  (b) Precision (effectiveness measure: g-mean)

(c) Specificity (effectiveness measure: g-mean)

Figure 5: **Threshold Optimization sets comparison, with g-mean as the effectiveness measure** – Critical distance diagram showing the results of the Nemenyi test. The horizontal axis represents the average rank of the strategies. A black bar connecting two or more strategies indicates no significant difference.

A.3   ADDITIONAL INFORMATION ON THE EXAMPLE DISCUSSED IN SECTION 6

Section 6 aims at discussing a qualitative example to illustrate and better understand the results obtained from our experimental analysis. For clear visualization, we selected a scenario that exhibits good separability (AUROC > 0.8 on the Evaluation set), and that shows the maximum performance variability among strategies. Consequently, the selected scenario is composed of the Mahalanobis monitor, used with the Resnet NN on the CIFAR10 ID dataset, and with the FGSM attack as the threat.

Tables 5 and 6 show additional information about this example. More specifically, we present the values taken by the five evaluation metrics on the Threshold Evaluation set, as well as the AUROC score for each threshold optimization strategy (ID, ID+T, ID+O), and each effectiveness measure.

Table 5: **Monitoring performances for the selected qualitative example, with OF+F1 as the effectiveness measure** – Measured metrics scores on the Threshold Evaluation set with different strategies, with OS+F1 as the effectiveness measure.

| Strategy | F1 | g-mean | recall | precision | specificity | AUROC |
|----------|-------|--------|--------|-----------|-------------|-------|
| ID       | 0.587 | 0.730  | 0.971  | 0.421     | 0.549       | 0.848 |
| ID+T     | 0.636 | 0.787  | 0.909  | 0.490     | 0.681       | 0.848 |
| ID+O     | 0.629 | 0.780  | 0.937  | 0.473     | 0.649       | 0.848 |

Table 6: **Monitoring performances for the selected qualitative example, with OF+F1 as the effectiveness measure** – Measured metrics scores on the Threshold Evaluation set with different strategies, with g-mean as the effectiveness measure.

| Strategy | F1 | g-mean | recall | precision | specificity | AUROC |
|----------|-------|--------|--------|-----------|-------------|-------|
| ID       | 0.636 | 0.787  | 0.909  | 0.490     | 0.681       | 0.848 |
| ID+T     | 0.643 | 0.791  | 0.879  | 0.507     | 0.713       | 0.848 |
| ID+O     | 0.589 | 0.719  | 0.613  | 0.568     | 0.843       | 0.848 |

