# OpenReview forum: "Red Pill or Blue Pill? Thresholding Strategies for Neural Network Monitoring"
_ICLR.cc/2024/Conference — Submitted to ICLR 2024_

### Official Review · Reviewer_qZZ4 · 2023-10-30

**Soundness:** 1 poor
**Presentation:** 1 poor
**Contribution:** 1 poor
**Rating:** 1
**Confidence:** 3

**Summary:**

In this paper, the authors analyze the problem of rejection threshold tuning, especially focusing on the available data with which the optimal threshold can be estimated.

**Strengths:**

This work addresses a crucial topic regarding the deployment of neural networks in critical applications, which encompasses several fields linked to ML robustness (OOD, uncertainty quantification, reject-based classification, etc.): although it is well-known in the research community it is also far from being solved.

**Weaknesses:**

The contribution of this paper is unclear.
Threshold-based rejection has been studied for decades since the seminal work of Chow in 1970 - and many of them are not considered in this paper. Several techniques to apply the rejection, estimate an optimal threshold, and evaluate the performance of classifiers have been proposed. The suitability of these methods strongly depends on the classification task, as every domain and particular application has its own requirements and thus different costs for rejects and misclassifications. For this reason, it's very hard to establish a general rule to perform this task.
Moreover, to the best of my knowledge, all the previous works perform the threshold tuning on a validation set from the same distribution of the training set, adding in some cases synthetic data (for instance through data augmentation). The knowledge of the "threat" is not considered, as it does not represent a realistic setting.
The obtained results are not surprising: if the rejection mechanism is "fitted" to reject data from a certain distribution, the performance is better. Adding other unrelated data makes the rejection of the considered threat harder, as the reject regions will be more complex - but it has to be considered that the classifier will be able to reject data from different distributions.

**Questions:**

- Can you please add in the paper more details to help the reproducibility of the experimental results? For instance, the details on how the threat datasets are constructed are not complete (e.g. adversarial attacks and their parameters).
- The experiments consider only the image domain, can you please extend them to other classification tasks?

---

### Official Review · Reviewer_f9sx · 2023-11-06

**Soundness:** 3 good
**Presentation:** 3 good
**Contribution:** 1 poor
**Rating:** 3
**Confidence:** 3

**Summary:**

This paper investigates threshold selection strategies for neural network monitoring tools, e.g., OOD detectors. Four different strategies for an optimization set are considered (ID-only, ID+Target, ID+Target+Other, and ID+Other). The threshold is selected to maximize either F1 score or g-mean metrics, and then evaluated on a number of additional metrics. Experiments show that ID-only

**Strengths:**

- The paper explores an important but understudied component of the OOD detection pipeline: selecting a threshold
- The experiments have wide coverage, spanning many different OOD detection methods and metrics

**Weaknesses:**

I'm not convinced by the basic premise of the evaluation setup: A metric is used on the optimization set to select a threshold, and this threshold is then evaluated against numerous different metrics. Isn't the point of threshold-independent metrics like AUROC and AUPR that different applications will have different tolerances for different kinds of error? This is widely recognized in binary classification and in the OOD detection research community, so it seems a bit odd to have thresholds be the unit of evaluation. It's useful to know whether thresholds that are good for one metric are also good for other metrics, but there is still a fundamental ambiguity about the requirements of the problem. There is a substantial literature on cost-sensitive learning that goes one step further in formalizing this, which would be good for the authors to discuss.

**Questions:**

How are thresholds selected in the ID-only case? That is, what are the positive and negative examples in the ID optimization set? I wasn't able to easily figure this out from the text.

---

### Official Review · Reviewer_VKDA · 2023-11-07

**Soundness:** 4 excellent
**Presentation:** 2 fair
**Contribution:** 2 fair
**Rating:** 5
**Confidence:** 2

**Summary:**

This paper considers threshold optimization of image classifiers. It identifies four different data sets to train threshold optimization methods: one data set with in-distribution data, and three data sets with different combinations of "threat data".  Threat data This is to answer two questions, namely 1) whether it is helpful to have prior knowledge about threats encountered at test time, and 2) whether it is helpful to optimize the threshold with threats that are not

**Strengths:**

1. This paper considers an important problem, and is highly relevant to present industrial and academic research.
2. This paper identifies two findings which can be useful to future research.
3. The empirical study covers multiple architectures, image data sets, different monitoring approaches, .

**Weaknesses:**

1. The main contribution of this paper are findings from an empirical study on what type of threat data is useful when optimizing the classification threshold. If the focus of a paper are empirical findings, the experimental design should be motivated more thoroughly and any findings should be phrased by mentioning the limitations of the experimental design.
2. The introduction and abstract of the paper are formulated quite generally. The contribution should be limited and results should be mentioned in more detail.
3. For me the focus of the paper, threshold search for runtime monitors, could guide the first four pages of the paper more. The authors could describe in more detail what they mean by monitoring and what differentiates the monitoring techniques they chose to assess, and why they chose them. Similarly the choice of threats could be motivated and explained in more detail.
4. The presentation in the first four pages obscures simple concepts and leaves many important questions unanswered. For instance, I wonder what the relation is between $f$ and $m_f$ and further what threat data can entail. Table 1 took me a long time to understand whereas it only says that the test data only contains target threat data that was not used in the training data set and no other generic threat data.
5. The experimental study is not repeated over multiple seeds. It is not clear how high the variance of the results are.
6. The choice of aggregating the results over multiple experimental settings might obscure findings. There are other ways to present complex experimental results. First, the non-aggregated results could be presented in the appendix. Second, plots with differently colored and shaped markers could be used.

**Questions:**

1. What do you mean by 'For training, there is no conceptual distinction between OOD and OMS'?
2. Can the authors find a better word for “OODness”, maybe just OOD or "what constitutes as OOD"?
3. What is the difference between $f$ and $m_f$?
4. In the work of Arampatzis & van Hameran (2001), does the effectiveness measure depend on the ground-truth labels of the test data?
5. Weaknesses of the analysed approaches were not discussed. Can the authors think of any?
6. Paragraph headings should be in small caps and also the usage of an em dash for additional separation is not necessary.
7. Why positive and negative ratio of 0.4 and 0.6?

---

### Meta-Review · Area_Chair_sUiF · 2023-11-29

**Metareview:**

The paper investigates threshold selection strategies for neural network monitoring. The problem is important in practice and the empirical study is extensive. However, there are concerns raised by the reviewers on the evaluation setup, contribution of the paper compared to prior work, and unclear writing, which are unanswered by the authors. Therefore, I recommend a rejection.

**Justification For Why Not Higher Score:**

There are concerns raised by the reviewers on the evaluation setup, contribution of the paper compared to prior work, and unclear writing. The authors didn’t engage with the reviewers to address their concerns.

**Justification For Why Not Lower Score:**

N/A

---

### Decision · Program_Chairs · 2024-01-16

Reject